# Microbiological Activity Affects Post-Harvest Quality of Cocoa (*Theobroma cacao* L.) Beans

Edy Subroto *, Mohamad Djali, Rossi Indiarto, Elazmanawati Lembong and Nur Baiti

Department of Food Industrial Technology, Faculty of Agro-Industrial Technology, Universitas Padjadjaran, Bandung 45363, Indonesia; djali@unpad.ac.id (M.D.); rossi.indiarto@unpad.ac.id (R.I.); elazmanawati.lembong@unpad.ac.id (E.L.); nur17003@mail.unpad.ac.id (N.B.)
* Correspondence: edy.subroto@unpad.ac.id

**Abstract:** Cocoa beans are the basic ingredient to produce chocolate and its derivatives, including cosmetics, foods, and pharmaceutical products. The quality of cocoa beans is greatly affected by post-harvest handling, especially by microbial activity involved in pre-conditioning after they are harvested, including fermentation, drying, and storage. This review aims to provide various factors that affect each stage of post-harvest cocoa beans, process mechanisms, and various latest technologies that can be used to improve the quality of cocoa beans. Microorganisms could be involved in each post-harvest stage and affect the cocoa beans' quality. However, fermentation was one of the keys to determining the quality of cocoa beans because fermentation involved various microorganisms, such as yeast, lactic acid bacteria, and acetic acid bacteria, which were interrelated primarily to produce precursor flavor compounds. The drying and storage processes were decisive in maintaining quality, especially in preventing mold growth and other microbial contaminants. Various technologies could improve the quality of cocoa beans during post harvest, especially by adding microbial starters during fermentation. Using several technologies of vacuum drying and a controlled atmosphere during storage could maintain the quality of the cocoa beans. However, many challenges must be faced, especially those related to controlling microbial activity during post-harvest. Therefore, post-harvest technology needs to be continuously developed, especially in controlling microbiological activities to improve the quality of cocoa beans effectively.

**Keywords:** cocoa beans; microorganism; post harvest; quality; fermentation

## 1. Introduction

The quality of cocoa beans depends on several factors, including the variety or type of cocoa, the growing environment, and the post-harvest handling of the cocoa beans [1–3]. Proper post-harvest handling can improve quality and prevent product loss. Post-harvest of cocoa beans includes sorting, ripening the fruit, breaking/splitting the fruit, fermenting, drying, and storage [4–6]. Each post-harvest stage can involve both beneficial and detrimental microbial activity. Therefore, controlling the factors and microbial activity during post-harvest handling is necessary. The important factor that must be considered, especially in fermenting cocoa beans. The method or conditions of the fermentation process that is carried out determines the final quality, especially in the formation of flavor precursors in cocoa beans. Fermentation can also reduce bitterness, change the color of the beans to black-brown, and change the texture of the beans hardens like a shell [7–10].

The low quality can be caused by the cocoa beans not being fermented properly or intentionally not fermented. Some farmers only soak fresh cocoa beans using water, which aims to remove the pulp, and then is immediately carried out the drying process without any fermentation process, even though fermentation is an important aspect of producing good quality cocoa beans [7,11,12]. The fermentation process involves the activity of many microorganisms. However, fermentation involves beneficial and harmful microorganisms [13,14]. The role of beneficial microorganisms is to assist in the fermentation

process by breaking down carbohydrates into simple compounds in the form of flavor precursors. In contrast, harmful microorganisms contaminate and can make cocoa beans rot or spoil [15,16].

The fermentation of cocoa beans occurs naturally with the help of microbes and lasts 6 days [17,18]. Microorganisms in fermentation will produce ethanol, acetic acid, and lactic acid compounds. Acetic acid and alcohol diffuse into the interior of the cocoa bean, followed by an increase in temperature, causing the seeds to die [18–20]. The presence of *Lactobacillus lactis* and *Acetobacter aceti* can shorten fermentation time [21,22]. Microorganisms that have a role in the early stages of fermentation are dominated by yeast, then followed by the growth of lactic acid bacteria, and then ended by the growth of acetic acid bacteria [7,23–25]. The mold growth during the fermentation needs to be avoided because the mold can produce mycotoxins and cause cocoa to have a bitter taste [21,26].

There are two methods of post-harvest processing of cocoa beans: dry cocoa beans with fermentation and dry cocoa beans without fermentation [27]. Non-fermented dry cocoa beans produce low-quality products. The quality of the non-fermented dry cocoa beans can be improved by the dry bean fermentation method. Still, the fermentation that is carried out must be optimized and conditioned as well as possible so that it can run well [12,28]. Various technologies have been developed to improve the quality of cocoa beans, especially by utilizing and controlling the role of microorganisms during post harvest.

Therefore, this review discusses in more depth the activities of microorganisms involved in the post-harvest handling process of cocoa beans. This review also discusses the effects of the activities of these microorganisms on the quality, as well as efforts to improve the quality of cocoa beans, especially by adding starter culture in the fermentation process and maintaining the quality of cocoa beans by applying drying and storage technology.

## 2. Microbiological Activity in Post-Harvest Handling of Cocoa Beans

Cocoa beans were essentially sterile while still inside the fruit, but the pulp often became contaminated by various microorganisms after being exposed. Acetic acid bacteria, yeast, lactic acid bacteria, as well as molds were commonly found in commercial cocoa beans [6]. Furthermore, unwanted contamination could occur during the processing and transportation of these beans [26,29]. Therefore, each stage in the post-harvest of cocoa beans requires process control, both microbiological and environmental controls. The stages of post harvest of cocoa beans and the factors that need to be considered schematically can be seen in Figure 1.

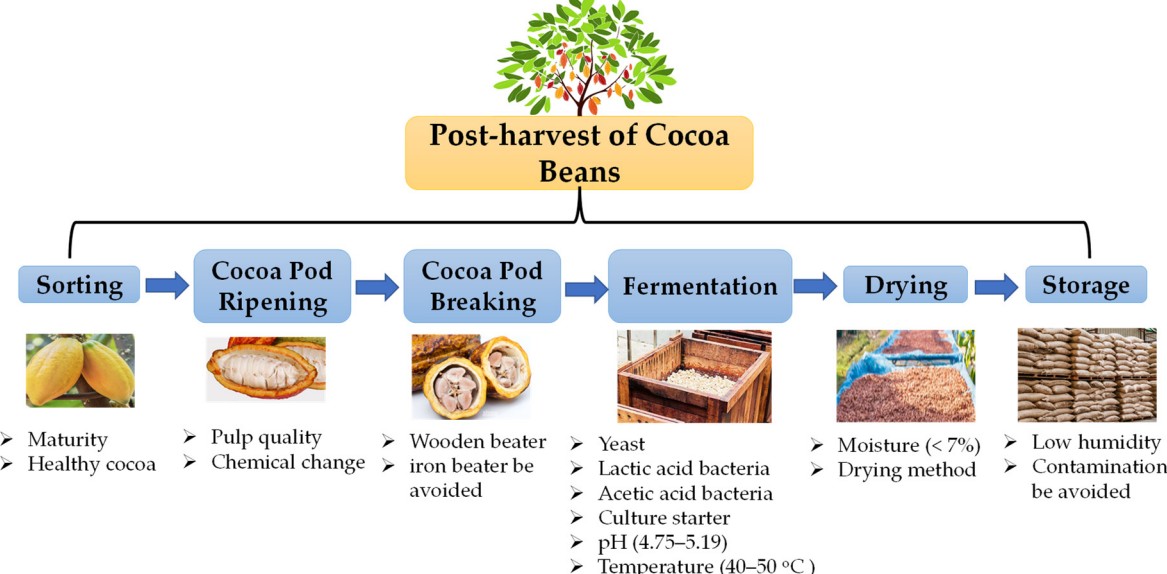

**Figure 1.** The stages of post harvest of cocoa beans and the factors that need to be considered.

Post-harvest handling is a crucial problem that must be addressed, as it greatly affected the final yield of cocoa beans produced. Therefore, a chain strategy for post-harvest handling was needed, where the potential for the emergence of *Salmonella* and mycotoxins in cocoa derivative products was the major focus, as followed in the HACCP, as well as in microbiological quality guidelines issued [29,30]. According to Lima et al. [6], bacteria from the genus *Bacillus* were the main microorganisms found in this plant. Some of these *Bacillus* species could form spores that were highly resistant to heat, thereby surviving the roasting process [31,32].

Based on the conditions required for the cocoa beans fermentation process, microbial ecological studies focused on the constituent mycotoxins and the derivative products [33]. Other studies had also focused on the viability of *Salmonella* during the handling of cocoa beans and derivative products [34,35]. These studies were not only relevant in terms of quality and safety but also provided an opportunity to identify microorganisms with new physiological characteristics [16,36].

Potentially, pathogenic microorganisms could also grow on this plant, such as *Aspergillus*, *Penicillium*, and *Fusarium*. These fungi often posed a risk to public health due to their ability to produce toxins [37]. Furthermore, their existence had the potential to cause mycotoxin contamination, which could endanger human health. The types of mycotoxins often found in cocoa beans included aflatoxins and ochratoxins, which were produced from *Aspergillus* and *Penicillium* species [34,38].

Cocoa could be the origin of chocolate contamination by bacteria, such as *Salmonella* due to poor hygiene practices during the processing [39]. However, mycotoxin-producing molds were a prominent concern in post-harvest handling because they determine the quality of the product cocoa beans. The optimal temperature of *Aspergillus* sp. for the formation of aflatoxin was 5 °C–45 °C with 80% minimum humidity and 5.5–7.0 pH [40]. Several types of mold that commonly contaminate cocoa included *Aspergillus flavus*, *Aspergillus niger*, and *Penicillium* [33,34,38].

Contamination from mold could cause weathering, reduced nutrition, and the presence of mycotoxins, thereby causing health problems [33]. The growth of these fungi was due to hydration in foodstuffs, problems during transportation, and poor storage [41]. The existence of toxins contained in cocoa beans was very detrimental because it led to a decrease in quality and could interfere with the health of consumers [42,43].

Therefore, several factors that influenced microbes in the post harvest of cocoa beans must be considered, including (1) cocoa genotype, which will affect the type and amount of polyphenols, carbohydrates, and proteins that were used as substrates by microbes during post harvest; (2) pod storage, which reduced pulp volume and increased micro-aeration which then reduced the formation of acetic acid and alcohol during fermentation; (3) de-pulping which reduced pulp sugar content which then reduced acid production; (4) fermentation conditions, which included cleanliness of equipment, environment, the addition of starter, reversal of cocoa beans, and length of fermentation; (5) drying, which included weather, technique, and drying equipment; and (6) microbial contamination, especially pathogenic bacteria and fungi, which produced harmful toxins.

### 2.1. Sorting

Harvested cocoa pods must be sorted to select good quality pods because the initial condition or quality determined the process conditions at later stages. Sorting determined the uniformity of the process in the next stage, namely, ripening. Furthermore, excessively ripe, damaged, or diseased cocoa pods could not be ripened because it often caused rotting and contamination [44,45]. The sorting process was often performed to separate healthy cocoa pods from the damaged variant [46].

This process must also be carried out to determine the quality of the product by separating cocoa beans from the impurities, followed by grouping based on quality and physical appearance [47]. The aspects that need to be considered during sorting include clipped, germinated, raw, flat, unfermented, purple, broken, dirt, insect, and moldy beans [48].

The moldy type was caused by wet cocoa beans that germinated and became dry, but the product overgrew, leading to the appearance of mold [49].

The criteria often considered during the sorting stage included size, color, shape, and health. The size of cocoa beans could be divided into 3 classes: large, medium, and small. Meanwhile, the color was typically distinguished by brown, purple, and black. The shape of cocoa beans could either be oval, round, or flat. Healthy beans were also separated from unhealthy, defective, or imperfect types [50,51].

*2.2. Cocoa Pod Ripening*

The ripening of cocoa pods typically involved storage for a certain period before breaking. Furthermore, this process had a significant effect on the product in terms of the chemical content and was useful for developing the flavor [52,53]. Ripening aimed to facilitate the formation of empty spaces in cocoa heaps, thereby aiding the penetration of oxygen during fermentation. The process also helped in reducing the pulp layer, speeding up fermentation, as well as decreasing water content, acidity, and polyphenols levels. Cocoa pods with poor quality, such as overripe, damaged, and diseased types could not be ripened due to rotting and contamination [54].

Changes in the properties of the pulp could occur during the ripening process of cocoa pods, including a reduction of the volume of pulp per bean due to water evaporation and conversion of sucrose, a decrease in the total sugar content, an increase in micro-aeration in the pulp, and a reduction of the production of alcohol and acetic acid [2]. The presence of excess seed pulp could affect the fermentation process, leading to a sour taste. Based on previous studies, products with low pulp volume and water content often experienced faster fermentation compared to others [55].

The process of ripening cocoa pods could be carried out In an open room or directly in the garden to prevent the proliferation of mold [48]. Hamdouche et al. [56] reported that several microorganisms were found in products aged 8 days, including *Acinetobacter* sp., *Klebsiella pneumoniae*, and *Bacillus* spp. These bacteria commonly proliferate on the surface during the ripening stage. *Acinetobacter* sp. caused nosocomial infections, which were widely spread in health facilities and could be transmitted through contaminated objects, touch, and saliva. These infections often occurred in the intestine and urinary tract, with several symptoms, such as diarrhea, fever, abdominal cramps, and lower abdominal pain. Several studies had shown that *Acinetobacter* sp. could grow on almost all types of surfaces, including cocoa pods [57,58]. *Klebsiella pneumoniae* was bacteria known for its ability to produce enzymes, leading to resistance to antibiotics. *Klebsiella pneumoniae* caused nosocomial infections, such as *Acinetobacter* sp. [58]. According to a previous study, it also caused pneumonia, which began with symptoms of fever, dry cough, and malaise [59]. Meanwhile, *Bacillus* spp. was a plant-grown promotion *rhizobacteria (PGPR)* bacteria, which could increase growth and production in plants. This bacteria had been reported to have the possibility of causing endocarditis and endophthalmitis [60]. The aforementioned bacteria had been used as biocontrol agents, phosphate solvents, nitrogen fixatives, and phytopathogens [61,62].

The time required for the ripening process of cocoa pods could significantly affect the organic matter and moisture content. The longer the time used for ripening, the greater the percentage of organic matter and the lower the water content [54].

*2.3. Cocoa Pods Breaking*

Fruit breaking referred to the process of removing and separating cocoa beans from shells. The breaking was often carried out using tools, such as sickles and paddles, by hitting the pods with each other or with a de-podding machine [63,64]. Furthermore, the cracking process was rarely performed mechanically because the tools used were usually not developed into commercial tools, and it was difficult to separate the fresh cocoa beans from the broken shells [65].

The best breaking method involved the use of a wooden beater. Cracking cocoa pods using an iron beater must be avoided because it could lead to black coloration, iron smell, as well as reduced aroma and taste. Meanwhile, the use of iron cutting tools could cause an oxidation reaction of phenolic compounds in cocoa beans, and the cut scars were easily overgrown by fungi [65]. The existence of mold on cocoa beans was not allowed because it reduced the quality of the product [14,49].

*2.4. Fermentation*

Fermentation referred to activity carried out by microorganisms either aerobically or anaerobically, which could cause changes in complex compounds into simpler types. Furthermore, the success of fermentation depended on the presence of microorganisms, and each of these microbes had different living conditions, such as pH, temperature, humidity, substrate, oxygen content, and others [16,66]. This process involved the decomposition of sugar components and citric acid compounds present in the pulp into organic acids by microbes [67]. The microbial activity present during this process facilitated the formation of flavor precursor compounds by changing the carbohydrate substrates into acetic acid, lactic acid, and ethanol [68]. Fermentation of cocoa beans was generally carried out for 5–6 days, and the first inversion treatment was conducted on the 2nd day or after 48 h, with further inversion being performed once every 24 h [69]. Fermentation occurred naturally without the addition of a starter culture due to the presence of glucose, sucrose, citric acid, and fructose in the pulp, leading to the growth of microorganisms [70,71].

The pulp of fresh cocoa beans contained water (80–90%), sugar (10–15%), and protein (0.5–0.7%). This indicated that it was an important source of macronutrients for microbial growth [7]. Furthermore, the water activity (Aw) of the pulp ranged from 0.98 to 0.99 and these conditions supported the spontaneous growth of microorganisms to carry out fermentation, such as lactic acid bacteria (LAB) and acetic acid bacteria (AAB) [72]. The pulp was very suitable for the growth of microorganisms, and during the fermentation process, the activity of microbes led to the production of organic acids and alcohol, as well as the release of heat (exothermic reaction) [73]. The exothermal reaction caused the diffusion of metabolites into the beans, leading to their death. This was followed by enzymatic reactions to form flavors, colors, and aromas that determined the quality of cocoa beans [74].

Cocoa bean fermentation was assisted by several types of microorganisms, namely, yeast, lactic acid bacteria, and acetic acid bacteria [12,14]. Yeast was pioneering microorganisms in the process of fermenting cocoa beans. This microbe could facilitate the production of proto-pectinase enzymes, leading to the breakdown of pectin compounds into pectin acids and alcohols, and pectinase enzymes. Subsequently, the pectinase enzyme catalyzed the breakdown of pectin acids into arabinose, galactose, and acetic acid [7,14]. The pulp was then crushed and released due to the decomposition of pectin [25]. Yeast and lactic acid bacteria were microorganisms involved in the decomposition of citric acid during the fermentation of cocoa beans [14,75].

The main aim of fermentation was to remove the mucus and kill cocoa beans, leading to the occurrence of changes, such as the formation of flavor and color precursors [69,76]. Factors affecting this process included the time or length of the procedure, aeration, fruit ripeness, quantity, uniformity of the speed of turning/stirring, climate, and the container [77]. An extended duration could increase the number of germinating and moldy cocoa beans, while a short time led to the emergence of slaty samples (unfermented beans) [27].

The stages of fermenting cocoa beans involved the opening of the pods, followed by the removal of beans still covered with pulp, and collection in a container [69,77,78]. Fermentation of cocoa beans was carried out in aerobic and anaerobic [77,79]. The presence of citric acid compounds made the pulp environment acidic and initiated the growth of yeast, followed by the occurrence of the anaerobic process. Furthermore, aerobic fermentation was carried out by acetic acid bacteria and lactic acid bacteria. Fermentation also involved enzyme activity, including endoproteases, carboxypeptidases, aminopeptidases,

glycosidases, polyphenol oxidases, and invertases [80]. The process produced good quality cocoa beans, which was indicated by the color of beans being changed from brown to slightly purple brown, slightly crumbly, bitter, and slightly astringent taste, and a strong chocolate odor [47].

The fermentation was often initiated by the growth of yeast colonies, followed by lactic acid bacteria, acetic acid bacteria, and mold [20,56,81]. However, bacterial succession in cocoa bean fermentation can be affected by the type of yeast and filamentous fungi present. The succession led to the growth of *Enterobacteriaceae*, LAB, and AAB, which can generate a wide range of genetic metabolic potentials related to carbohydrate and protein metabolism. Based on in silico evidence, the interspecific quorum sensing (QS) arsenal found the genera *Bacillus*, *Lactobacillus*, *Enterobacter*, and *Pantoea*, while the potential for intraspecific QS was found in the genera *Lactobacillus*, *Komagataeibacter*, *Enterobacter*, *Bacillus*, and *Pantoea*. On the other hand, quorum quenching potential (QQ) was detected in the *Lactobacillus* and AAB groups. These QS and QQ can modulate the dominance of bacteria during cocoa bean fermentation, which is also affected by cross-feeding, over a long period of time [82].

Microorganisms in the fermentation process caused several biochemical changes, including the hydrolysis of sucrose in the pulp into fructose and glucose during the first 24 h [7,19,83]. The growth of microbes on this sugar caused an increase in temperature. The pulp had a high sugar content of 10–15% sugar, thereby stimulating the growth of yeast, which could convert these molecules into alcoholic compounds under anaerobic conditions and hydrolyze pectin compounds covering cocoa beans. The process was then accompanied by the death of yeast due to the presence of these alcohol compounds, leading to a higher temperature change. Furthermore, *Streptococcus* and *Lactobacillus lactic* acid bacteria were able to grow, and then the pulp was stirred for aeration purposes. The presence of oxygen and low pH caused acetic acid bacteria (*Acetobacter* and *Gluconobacter*) to grow [13,84]. The main stages of cocoa bean fermentation involving yeast, lactic acid bacteria, and acetic acid bacteria are presented schematically in Figure 2.

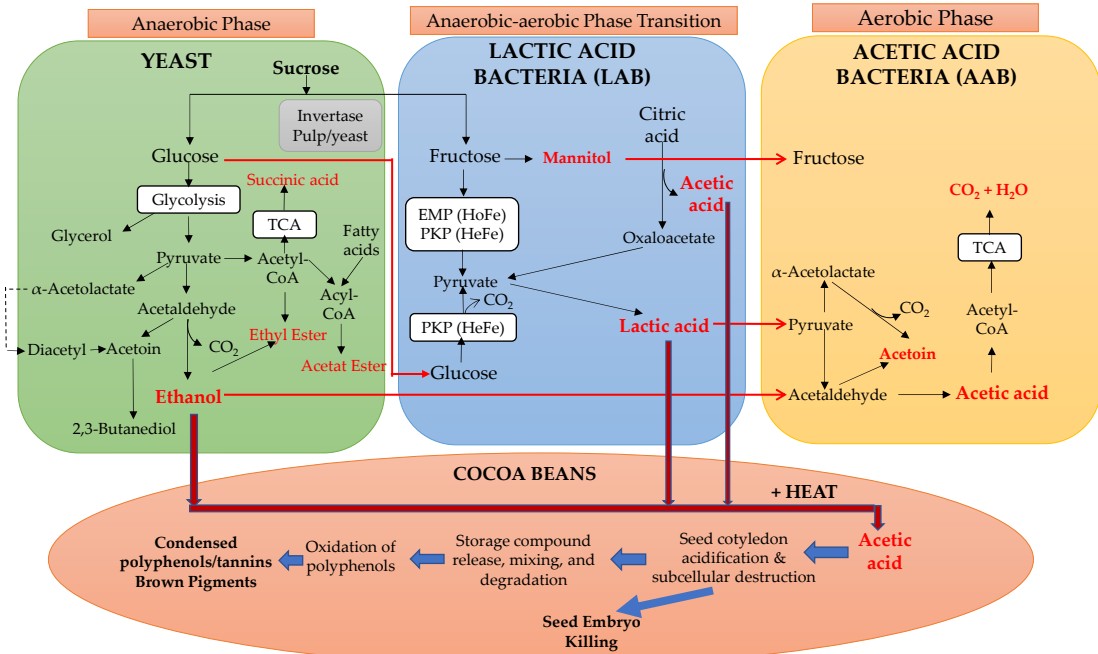

**Figure 2.** The main stages of cocoa beans' fermentation mechanism involve yeast in the anaerobic phase, lactic acid bacteria in the anaerobic–aerobic phase transition, and acetic acid bacteria in the aerobic phase.

During cocoa fermentation, there were approximately 100 million microbes per gram after 5–6 days. The process was then stopped and cocoa beans were dried because the

continuation of fermentation could cause an unwanted odor due to the growth of molds (*Aspergillus*, *Mucor*, and *Penicillium*), which hydrolyzed lipids [27]. Furthermore, some yeasts that grew earlier played a role in degrading the pulp and converting sugars into alcohol compounds. The process was carried out exothermically, with heat being generated, leading to an increase in the temperature of the cocoa mass that was being fermented [19,85]. The types of yeast involved could differ in each region, as shown in Table 1.

**Table 1.** Several types of yeast are involved in fermentation process of cocoa beans in several countries.

| Country | Yeast Species | Characteristics | References |
|---|---|---|---|
| Ecuador | *S. cerevisiae, R. minuta, P. manshurica, P. kudriavzevii, P. kluyveri, K. marxianus, H. opuntiae, C. tropicalis, C. sorbosivorans*-like, and *T. delbrueckii* | *P. manshurica, P. kudriavzevii,* and *S. cerevisiae* were the dominant ethanol producers. | [86] |
| Brazil | *P. kundriavzevii, C. orthopsilosis, K. ohmeri, D. etchellsii, I. orientalis, H. uvarum, P. kluyveri,* and *S. cerevisiae* <br> *P. Kluyver, C. magnoliae,* and *S. cerevisiae* | These species adapted well to both fermentation box and the stainless steel used in large-scale fermentation. | [87] <br><br> [76] |
| Cuba | *T. delbruekii, P. terricola, C. ortopsilosis, P. occidentalis, C. tropicalis, P. kluyveri, P. kundriavzevii, H. opuntiae,* and *P. manshurica* | *Pichia kudriavzevii* was the most common. Some yeasts undergo mutations caused by natural processes, such as transposons genetic recombination, changes in ploidy, and sexual reproduction. | [67,88] |
| Dominican Republic | *C. zeylanoides, Y. lipolytica, H. guillermondii,* and *C. inconspicua* | *Candida inconspicua* was the most common and dominant because it could survive up to the 36th hour of fermentation. | [89] |
| Indonesia | *S. Cerevisiae, Kloeckera* sp., *S. fibuligera, C. tropicalis,* and *C. krusei* <br> *C. tropicalis, S. cerevisiae,* and *Kl. apis* | Yeast found in Indonesia could generally live in tropical environments. *Saccharomyces cerevisiae* and *Candida tropicalis* were resistant to high temperatures (>40 °C). | [90] <br><br> [25] |
| Ivory Coast | *G. geotrichum, W. anomalus, P. galeiforms, P. kudriavzevii, C. tropicalis, S. cerevisiae, P. kluyveri,* and *P. kundriavzevii* <br> *P. fermentans, P. klyvera, Candida* sp., *C. insectorum, P. kudriavezii, I. hanoiensis, P. sporocuriosa, P. manshurica,* and *H. opuntiae* <br> *P. kudriavezii, I. hanoiensis, P. sporocuriosa, P. manshurica,* and *H. opuntiae* | *Pichia kudriavzevii, Pichia kluyveri,* and *Saccharomyces cerevisiae* were the most common and had intraspecific diversity. | [91] <br><br><br> [92] <br><br><br> [56] |
| Ghana | *H. guilliermondii, P. membranifaciens, Sc. cerevisiae, S. crataegensis, P. Pijperi, I. Hanoiensis, C. zemplinina, C. michaelii, C. diversa, C. ethanolica, Schiz. pombe,* and *I. orientalis* | *H. guilliermondii* was the most common species at the beginning of fermentation (0–24 h), while *P. membranifaciens* was the dominant species at the end of fermentation (36–144 h). | [70] |

**Table 1.** *Cont.*

| Country | Yeast Species | Characteristics | References |
|---|---|---|---|
| | *P. manshurica M.(P.) carribica, K. ohmeri, C. orthopsilosis, C. carpophila, H. opuntiae, S. cerevisiae, P. kundriavzevii* | *S. cerevisiae* and *P. kundriavzevii* were the most common species. *Hanseniaspora opuntiae* was able to live at a fairly low pH. | [93] |
| | *Saccharomyces cerevisiae, Kluyveromyces lactis, Candida glabrata,* | *S. cerevisiae* and *K. lactis* were the most common species. | [13] |
| Mexico | *H. guilliermondii, S. crataegensis, S. cerevisiae,* and *P. kundriavzevii* | *S. cerevisiae* was the most important species because it had the best survival. | [94] |

The fermentation process was followed by the growth of several types of bacteria. Furthermore, bacteria belonging to the Beta genus, which were heterofermentative and involved in cocoa bean fermentation, were capable of producing acetic acid and lactic acid [95,96]. Some acetic acid bacteria also played a role in the oxidation of alcohol compounds to acetic acid. The microbes involved in the fermentation of cocoa beans included lactic acid bacteria and acetic acid bacteria [97]. The types of acetic acid bacteria and lactic acid bacteria implicated in fermentation were different in each region [81,97,98]. The diversity of these microorganisms was caused by different factors in each region, such as oxygen levels, temperature, and fermentation techniques used [20,96]. The various species present in each country are presented in Table 2.

**Table 2.** Several types of bacteria species are involved in cocoa bean fermentation in several countries.

| Country | Bacterial Species | Characteristics | References |
|---|---|---|---|
| Ecuador | *Lb. fermentum, A. pasteurianus, Leu. pseudomesenteroides, Lb. plantarum, A. fabarum, F. tropaeoli*-like, *Lb fabifermentans, Lac. lactis, Lb. nagelii, Lb. cacaonum, E. casseliflavus, A. peroxydans, A. cibinongensis,* and *A. malorum/indonesiensis* | *Lb. fermentum, A. pasteurianus,* and *Leu. Pseudomesenteroides* were the most commonly found in fermented cocoa beans. | [86] |
| Indonesia | *B. licheniformis, B. pumilus, A. pasteurianus L. plantarum,* and *L. cellobiosus* | *L. plantarum* was the most consistent bacteria, while lactic acid bacteria had a dominant role in the microbial ecology of cocoa beans fermentation. | [25] |
| Ghana | *Ent. faecium, Ent. casseliflavus, W. ghanensis, Leuc. Mesenteroides, Leuc. pseudomesenteroides, L. mali, L. brevis, L. plantarum,* and *L. Fermentum* | *L. plantarum* was the most commonly found in fermented cocoa beans. *Weisella ghanensis* was known as the first-line divergent in the genus *Weisella*. | [99] |
| | *Lactiplantibacillus plantarum, Lactobacillus nagelii, Liquorilactobacillus cacaonum, Limosilactobacillus fermentum,* and *Leuconostoc pseudomesenteroides* | | [100] |
| | *Lb. plantarum, A. pasteurianus, Leu. mesenteroides, G. oxydans, G. diazotrophicus, G. hansenii, Leu. citreum, Lb. fermentum, Lb. brevis,* and *E. coli* | | [13] |

**Table 2.** *Cont.*

| Country | Bacterial Species | Characteristics | References |
|---|---|---|---|
| | *Lb. Plantarum, Pd. acidilactici, Lb. hilgardii, Lc. pseudoficulneum, Lb. fermentum, G. oxydans, A. malorum, A. tropicalis, A. syzygii,* and *A. pasteurianus* | | [70] |
| | *A. tropicalis*-like, *A. tropicalis, A. syzygii*-like, *A. senegalensis, A. senegalensis,* and *A. pasteurianus* | | [21] |
| Dominican Republic | *L. paracasei* subsp. *paracasei, L. brevis, L. pentosus,* and *L. plantarum* | *L. plantarum* was most commonly found in fermented cocoa beans. | [89] |
| Ivory Coast | *W. cibaria, W. paramesenteroide, L. casei, F. pseudoficulneus, Ent. faecium, L. curieae, Leuc. mesenteroides,* and *L. plantarum,* | *Leuconostoc mesenteroides* was most commonly found in fermented cocoa beans. *Leuconostoc mesenteroides* could catabolize citrate more efficiently. | [101] |
| | *A. malorum, A. ghanensis, A. okinawensis, A. tropicalis, A. pasteurianus,* and *G. oxydans* | *A. pasteurianus, A. okinawensis,* and *A. tropicalis* were the most commonly found acetic acid bacteria in fermented cocoa beans. *Lactobacilli* and *lactococci* could metabolize sucrose, fructose, and glucose during fermentation. | [96] |
| | *Lactobacilli* sp., *Lactococci* sp. | *Lactobacilli* strains were unable to metabolize citrate, while *lactococci* strains could use citrate as a carbon source. | [75] |
| Brazil | *Oenococcus oeni, P. acidilactici, S. salivarius, F. pseudoficulneus, Lc. mesenteroides, Lc. lactis, L. reuteri, L. amylovo-rus, P. dextrinicus, L. brevis, L. acidophilus, L. delbrueckii, L. lactis, L. rhamnosus, L. casei, L. fermentum,* and *L. plantarum.* | *L. plantarum* was the most commonly found in fermented cocoa beans. *L. plantarum* adapted well to cocoa ecosystem by responding to changes in ethanol concentration, temperature, and acid stress. | [92] |
| | *Acetobacter senegalensis, Bacillus subtilis, Limosilactobacillus fermentum, Brevundimonas, Pseudomonas,* and *Kozakia baliensis.* | | [100] |
| | *G. saccharivorans, G. xylinus, Ga. oxydans, A. peroxydans, A. cerevisiae, A. malorum, A. indonesiensis, A. fabarum, A. lovaniensis, A. senegalensis, A. ghanensis, A. pasteurianus,* and *A. aceti* | | [102] |
| Ecuador | *W. fabaria, W. cibaria, L. satsumensis, F. ficulneus, E. saccharolyticus, L. amylovorus, L. cacaonum, L. nagelii, Lc. lactis* subsp. *lactis, L. fabifermentans, F. tropaeoli*-like, *Leuc. pseudomesenteroides,* and *L. fermentum,* | *L. fermentum* was the dominant and widely studied lactic acid bacteria. *L. fermentum* lived at the beginning of fermentation of cocoa beans and could change citrate. Assimilation of citric acid increased the pH levels, thereby allowing the growth of less acid-fast lactic acid bacteria species, facilitating acetic acid bacteria growth, and optimizing the expression of some microbial activity, such as pectinolytic activity by yeast. | [86] |

Lactic acid bacteria were generally gram-positive bacteria and could be categorized into two groups, namely, homofermentative and heterofermentative. The homofermentative group only produced the final product in the form of lactic acid, while the heterofermentative group produced the final product of lactic acid and other compounds, such as acetaldehyde, carbon dioxide, and ethanol [103]. Furthermore, lactic acid bacteria had a crucial role in the fermentation of cocoa beans, especially in breaking down the sugar in the pulp through the homofermentative and heterofermentative pathways [81,104]. The activity of these bacteria produced acidic compounds, which diffused into beans, causing a decrease in the pH to a range of 4.0 to 5.0 [105].

Some of the lactic acid bacteria involved in cocoa bean fermentation are classified as probiotics, which are defined as "live microorganisms that, when administered in adequate amounts, confer a health benefit on the host" [106]. Therefore, the aspect of pro and postbiotics in fermented cocoa beans is one of the interesting potentials to be developed in cocoa and its processed products. One of the probiotics found in fermented cocoa beans is *Lactobacillus plantarum* [107]. Several other probiotics, such as *Bifidobacterium lactis*, *Lactobacillus rhamnosus*, *Lactobacillus acidophilus*, *Lactobacillus paracasei*, and *Lactobacillus casei* can also be added to the fermentation process or to some chocolate processing processes which can increase the probiotic content in the product [108,109]. *L. plantarum* is known to produce postbiotic metabolites in the form of bacteriocins, including RG14, RG11, RI11, RS5, TL1, and UL4, which have cytotoxicity capabilities against various types of cancer cells [110]. Foong et al. [107] reported that they succeeded in isolating *L. plantarum* from fermented cocoa beans and applying it to dark chocolate to produce a product that has the potential to be very good for health.

The growth activity of yeast, acetic acid bacteria, and lactic acid bacteria affected the concentration of lactic acid, acetic acid, and ethanol produced. *Saccharomyces cerevisiae* grew dominantly at the 24th hour of fermentation, followed by the optimum growth of *Lactobacillus lactis* at the 48th hour. Furthermore, *Acetobacter aceti* was already active at 24 h, but its maximum number was reached at 72 h [25]. Lactic acid bacteria converted sugar into lactic acid under anaerobic conditions, and acetic acid bacteria converted alcohol into acetic acid under aerobic conditions [111,112]. The activity of microorganisms implicated in the cocoa beans fermentation process and their roles can be seen in Table 3.

**Table 3.** Microorganisms' activity is implicated in cocoa bean fermentation.

| Microorganism | Lifetime | pH | Temperature | Role | References |
|---|---|---|---|---|---|
| Yeast | Lived at the beginning of fermentation, and then the population increased in the 24th hour. | 3.1–3.3 | 30–35 °C | Yeast converted glucose from the pulp into ethanol. Its decomposed pectin compounds into pectin acids and alcohols in the presence of proto-pectinase enzymes, then decomposed pectin acids into arabinose, galactose, and acetic acid using pectinase enzymes. Converted citric acid contained in the pulp. | [20,75] |
| Lactic acid bacteria | Grew from the beginning of fermentation, and then became dominant at 36 to 72 h. | 3.3–4.0 | 30–40 °C | Broke down sugar into lactic acid, pyruvate, and mannitol, and then lowered the pH. | [20,81,105] |
| Acetic acid bacteria | Grew from the beginning of fermentation, and then became dominant at 72 h. | 4.0–5.0 | 28–30 °C | Played a role in the process of oxidation of alcohol compounds (ethanol) to acetic acid. | [20,113] |
| Mold | Grew at moisture content > 8% | 2.0–8.5 | 25–30 °C | It caused the rotting of cocoa beans and produced toxins and other secondary metabolites. | [114,115] |

Contamination by other microorganisms, such as mold, was also often found during the fermentation of cocoa beans. Copetti et al. [33] reported that several molds were identified during the process for 6 days, including *Absidia corymbiera*, *Geotrichum candidum*,

*Penicillium paneum*, and *Monascus ruber*. The physiological aspects of molds caused them to be found in the fermentation of the products [14,114,116].

The presence of molds must be avoided due to their ability to cause a bitter taste and an off flavor in the product due to the production of several undesirable flavor compounds [14,116]. These microbes often proliferate due to contaminants from the surrounding environment. The presence of mold on the surface of cocoa beans' shells did not cause significant losses, but their penetration into the seeds caused damage to the color and flavor [117].

### 2.5. Drying

The drying process was a continuation of the oxidation stage of fermentation, which could reduce the bitterness and chelation of cocoa beans. Furthermore, drying produced dry cocoa beans with good quality in terms of physical characteristics, including strong flavor and aroma precursor. In conditions where the drying process was carried out slowly, this situation could be detrimental because it excited the growth of molds and their penetration into cocoa beans [118]. A relatively fast drying process could interfere with the oxidation reaction and cause excessive acidity. An increase in the temperature also caused an increment in acidity and chelation; hence, the maximum threshold was 70 °C [3]. Hayati et al. [119] reported that cocoa beans treated with fermentation and dried with 12 treatment combinations had the lowest water content obtained with a drying temperature of 60 °C for all treatments. This was due to the material's ability to release moisture from the surface and this was directly proportional to the increase in temperature.

Drying was often carried out after fermentation and was commonly useful for reducing the moisture content of cocoa beans to 6–8% [27]. Apart from using sunlight, this could also be conducted using a drying machine, such as an oven blower, especially when the weather was not sunny [120,121]. This alternative was utilized to prevent damage to cocoa beans. A moisture content of less than 6% could cause excessive brittleness, thereby complicating handling and further processing. Meanwhile, a content of >9% could cause weathering of beans due to the growth of fungi. Drying could be conducted by utilizing sunlight, which required approximately 10–14 days [3].

The drying process must only be carried out on cocoa beans that had already undergone fermentation. Freshly harvested cocoa beans often had a very high moisture content of approximately 51–60%, making them more prone to damage or rot due to the growth of microorganisms [27]. The drying was expected to reduce the water content in the product to around 7.5%, thereby preventing microorganisms from growing and prolonging the shelf life of cocoa beans. This process also made it easier to remove nibs from cocoa shells [3,27].

Salazar et al. [122] conducted observations of microbiological properties in post-harvest handling of cocoa beans, where 4 types of treatment were carried out, namely, treatment A and A-farmer, treatment B and B-farmer. Treatments A and A-farmer (performed by farmers directly in the field) were carried out by drying cocoa beans through an artificial process, while treatments B and B-farmers (carried out by farmers directly in the field) were performed using sunlight. The observation results showed that the highest total aerobic mesophilic (TAM) occurred in fresh samples, and *Salmonella* was not found in all treatments. The highest TAM and yeast values were also found in the fresh samples because cocoa beans in the pods were still sterile after being cut. The moisture and high sugar content in fresh cocoa beans were used for the growth and development of microorganisms [7,122].

The maximum allowable value of mold on cocoa beans in the Venezuelan standard was 3.00–4.00 log CFU/g [122,123]. Meanwhile, the international standard ISO 2451:2017 for the cocoa beans—specification and quality requirements have set a maximum limit for moldy dry cocoa beans, which is 3% for grade 1, and 4% for grade 2 [124]. Samples below this threshold still met the good manufacturing practice (GMP) standards. GMP standards had recommended a drying process for cocoa beans to avoid contamination, especially by fungi, including: (a) fermented cocoa beans were immediately dried using direct sunlight

or artificial methods until the moisture content was less than 8%; (b) the drying area must be protected from sources of contaminants and thickness of the stack of cocoa beans must be less than 6 cm; (c) the samples were turned over several times (5–10 times per day) to achieve total dryness; (d) cocoa beans must be protected from animals to avoid biological contamination; and (e) equipment must be cleaned regularly [125,126]. Microbes growing on dried cocoa beans could also vary, which was mainly influenced by the drying process and the conditions of the area where the samples were obtained. Furthermore, Delgado-Ospina et al. [127] stated that drying 18 samples of cocoa beans from different plantations led to the presence of some groups of microorganisms, such as yeast, mold, *Lactobacillus* spp., *Lactococcus* spp., total aerobic thermophiles, total aerobic mesophiles, and *Enterobacter* spp. The differences in microbes were caused by several factors, such as variations in the drying techniques used, the maturity level of the samples, storage, and environmental factors, including field conditions, weather, and the initial condition of cocoa beans [16].

Thermophilic bacteria could grow optimally at a high-temperature range of 40–80 °C. These microbes were also found in extreme environmental conditions, such as a pH of more than 10 or less than 2, high salinity (saturated NaCl), and substrate pressure [128]. Meanwhile, mesophilic bacteria could live at an optimum temperature range of 25–37 °C [129]. *Enterobacter* sp. had been reported to have various enzyme activities, including proteolytic properties, and it acted as an opportunistic pathogen [130]. One of the microbiological characteristics of the drying process of cocoa beans after fermentation is presented in Figure 3.

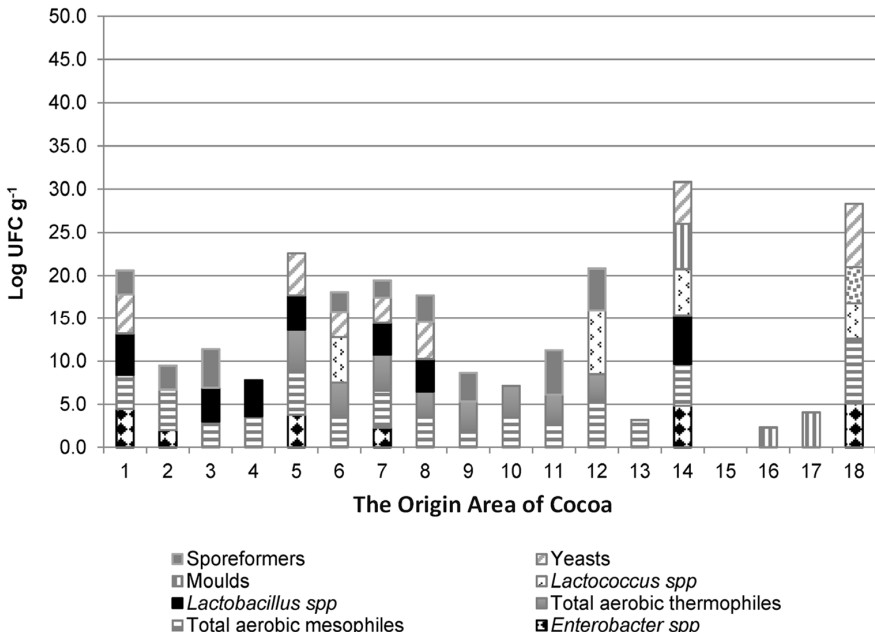

**Figure 3.** Microbiological characteristics of the drying process of Colombian Criollo cocoa beans (the origin area of cocoa, Valle del Cauca = 1−4, 6−8, 11−13, 15, 17, 18; Cauca = 5, 9, 10, 14; Narino = 16) [127].

Several species of mold were sometimes found during the drying of cocoa beans, such as *Absida corymbifera*, *Penicillium paneum*, *Aspergillus parasiticus*, *Aspergillus candidus*, *Aspergillus flavus*, *Aspergillus niger*, and *Eurotium chebalieri*. Furthermore, *Aspergillus flavus* had great potential as a producer of aflatoxins and ochratoxins [33,34,131]. *P. paneum* was a species that could grow in low oxygen conditions, 4–5 pH, and high $CO_2$ levels [132].

### 2.6. Storage

Dried cocoa beans must be stored under certain conditions to maintain their quality. During the storage period, several strains of mold could be found [116,133]. Copetti [33]

succeeded in identifying several strains, including *Aspergillus penicillioides*, *Eurotium rubrum*, *Eurotium chevalieri*, *Eurotium amstelodami*, and *Absida corymbifera*. These molds were a class of xerophilic fungi that could live in dry environmental conditions, aw, and low moisture content conditions. Furthermore, they were often found in dry products, such as spices, powdered herbs, and nuts. Xerophilic fungi had the potential to produce mycotoxins, which were harmful to health [134].

Based on previous studies, it was not advisable to store cocoa beans for a long time to avoid damage. This was because dry products were hygroscopic and could easily absorb moisture until they reach equilibrium conditions [135]. Under aw conditions and low moisture content, mold spores could survive for a long time. Poor storage, such as high humidity levels, caused spore germination, leading to rot and the presence of toxins. The relative humidity for storing cocoa beans was recommended to be around 68–70% or lower with a temperature of 25–30 °C [27,42,115]. Several types of fungi, such as *Aspergillus restrictus* and *Aspergillus glaucus*, could grow in environments with relative humidity of >70%. These two groups of fungi species were considered dominant in initiating damage to agricultural products, including cocoa beans. Samples with moisture of 8% or more were more likely to become moldy [27]. Sánchez-Hervás et al. [136] explored 9 samples of cocoa beans originating from Sierra Leone, Ecuador, and Guinea. The results showed that the dominant molds during the storage period included *Aspergillus flavus* and *Aspergillus niger*. Storage under controlled atmospheric conditions was also recommended because it could prevent deterioration due to environmental factors. The use of inert gases, such as carbon dioxide or nitrogen was also effective in maintaining the quality of stored agricultural products, including coffee and cocoa beans [137–140].

## 3. Effect of Microorganism Activity on Quality of Cocoa Beans

### 3.1. Physical Properties

The effect of microbiological activity on the quality of cocoa beans could be observed visually through the physical changes on the surface of the samples. Fermented samples had a rather dark brown color and a hollow texture. Physical assessment of the effect of microorganisms' activity on cocoa beans was assessed in more detail through a cut test. This split test was used as a standard for determining the degree of fermentation of the samples [55]. Furthermore, the principle of the test was that cocoa beans were split lengthwise using a knife to display the appearance of the cotyledon surface. The color of the two halves of beans was then observed visually, and the color was classified as gray (slaty), purple (violet), and fully brown for unfermented, partially fermented, and fully fermented samples [21]. Based on physical observation, slaty beans had a firm texture, did not produce a distinctive taste, had excessive astringent and bitter taste, low aroma quality (still smells of alcohol), and purple color, especially in Linda cocoa [141].

### 3.2. Chemical Properties

Chemical influences were also found on cocoa beans due to the activity of microorganisms in post-harvest handling, which included pH, total acid, ethanol content, and reduced sugar content [21]. Kouamé et al. [142] reported that during fermentation, there was an alteration in pH or acidity. The initial pH of cocoa beans before the process was 4.2, but decreased to 3.6 after 24 h. However, then there was an increase in indigo pH at the 60th hour to 5.8 and 7.3 at the end of the process.

Organic acids were formed and this occurred simultaneously with the production of ethanol by yeast at the beginning of fermentation during the anaerobic phase [7,143]. The ethanol produced was then used by lactic acid bacteria to produce lactic acid. Ethanol was produced at the beginning of the fermentation process by yeast due to the breakdown of sugar [144]. The most widely available reducing sugars included fructose and glucose, but there were also non-reducing sugars, such as sucrose. The sugar content, especially reducing sugars, decreased as fermentation progressed. This was because sugar was used as energy for physiological activity and seed metabolism [17].

The compounds produced during the process could be used as a determining parameter known as the fermentation index. This index was used as a benchmark for assessing the progress of fermentation in cocoa beans [145] and was also considered to have more objective results compared to the cut test. Furthermore, it compared the absorbance at a wavelength between 460 to 540 nm. Fermentation was considered good if the index produced was close to 1 [12].

The index value was determined based on the absorbance level of the fermented compounds and their formation [146,147]. Compounds that changed during the process included polyphenolic compounds, such as flavonoids. Based on previous studies, flavonoids could significantly decrease during fermentation up to 115–43 mg/g compared to non-fermented cocoa beans, which contained higher levels of these compounds [148–150]. Compounds produced from fermentation included tannin complexes that were brown in color and had maximum absorbance values at a wavelength of 460 nm. Some other compounds were reduced due to the process, including anthocyanins, with purple color and maximum absorbance values at a wavelength of 530. Potential flavors from the seeds of cocoa could be seen from the quality of its fermentation through fermentation index value [2].

Certain bacteria starters could also be added to fermented cocoa beans to facilitate the rate of the process. Kresnowati et al. [151] added *L. plantarum* as a starter and this led to an increase in the rate of sugar consumption by 64%, while its absence produced a 46% increment. The concentration of metabolites after the addition of the *L. Plantarum* starter culture also increased by 5–6 times for lactic acid, 4 times for acetic acid, and 50 times for ethanol compared to standard fermentation.

Fermented ethanol was useful for stopping seed germination and accelerating the death of cocoa beans. This was because the heat produced during the formation of ethanol did not damage the structure of the seed cells, thereby facilitating the diffusion of various substances out or into the seeds. *L. plantarum* was one of the microorganisms that produced ethanol in cocoa bean fermentation [152]. Furthermore, its addition was useful as a trigger to increase the growth rate of heterofermentative lactic acid bacteria and prevent its production by yeast. An increase in ethanol levels also affected the increment in organic acid levels in dry cocoa beans [9,71,152]. The pH value of cocoa beans' pulp tended to increase at the end of fermentation along with the extended duration. The activity of bacteria implicated in the process was generally a reaction that produced heat (exothermic) [12]. In spontaneously fermented cocoa beans, there was the growth of acidic bacteria, which showed a normal growth curve. This could be seen by the adaptation, exponential, stationary, optimum growth, and death phases at 0 to 4 h, 8 h, 8 to 12 h, 12 h, and 16 to 20 h, respectively [113].

## 4. Quality Improvement of Cocoa Beans

### 4.1. Post-Harvest Treatments for the Quality Improvement of Cocoa Beans

The relatively low quality of conventional cocoa beans could be improved using various methods. Quality improvement could be carried out starting from crop modeling to post-harvest handling of the product [5,153]. This was because post-harvest handling was still not optimal, thereby causing quality defects and non-compliance with the requirements [12,154]. Farmers with smallholder plantations did not often carry out fermentation properly, naturally, or with the addition of inoculum. These farmers only rinsed the samples using water and then dried them in the sun, leading to the brief occurrence of spontaneous fermentation or total absence [12,155].

The common method used to improve the quality of cocoa beans was through controlled fermentation [85,143]. The biochemical processes that occurred during the process using microbes produced several flavor precursors compounds. Furthermore, these compounds could improve or increase the quality of cocoa beans. Fermentation was carried out by several microbes, including lactic acid bacteria (*Lactobacillus* sp. *KSL2*), acetic acid bacteria (*Acetobacter* sp. *KLK1*), and yeast (*Candida* sp. *KLK4*) [16,19,20]. Various methods or post-harvest treatments to improve the quality of cocoa beans are presented in Table 4.

**Table 4.** Various methods or post-harvest treatments to improve the quality of cocoa beans.

| Post-Harvest Stage | Treatments | Conditions | Characteristics | References |
|---|---|---|---|---|
| Sorting and fermentation | Determining cocoa pod maturity and fermentation time to increase healthful bioactive compounds. | Pod harvest: mature and ripe. Post fermentation: 1, 3, 5 days. | Mature cocoa pods and fermentation for 3 days could produce cocoa beans with a high content of bioactive compounds, high antioxidant activity, and the desired flavor. | [52] |
| Fermentation | Fermentation by administering anti-fungal strains. | The anti-fungal strain of *L. fermentum* and *S. cerevisiae* were added to 180 kg box. | Culture of *L. fermentum 223* and *S. cerevisiae H290* had good anti-fungi activity and produced less off-flavor, a good percentage of fermented beans, less astringency, and the best cocoa taste. | [156] |
| Fermentation | Fermentation with the addition of a mixture of LAB and AAB starter cultures | Starter culture: *A. pasteurianus 386B, L. fermentum 222, S. cerevisiae H5S5K23* (3 heaps and 1 box). | The addition of lactic acid bacteria and acetic acid bacteria starter culture mixture accelerated carbohydrate fermentation by increasing the conversion of lactic acid and citric acid and was proven to improve the chocolate flavor produced. | [105] |
| Fermentation | Controlled temperature and pH during fermentation. | Temperature: 12–28 °C, 20–26 °C, and 16–28 °C. RH: 80–85%, 80–85%, and 70–75%. | Controlled fermentation of cocoa beans at a pH between 4.75 to 5.19 and a temperature below 40 °C could optimize activity of microbes and enzymes forming flavor compounds, as well as other sensory attributes. | [85] |
| Fermentation | Cocoa bean turning start times on fermentation | Cocoa beans: Criollo, turning start times: 24 and 48 h). | Turning start of 48 h could stimulate flavor-forming microbes, such as *M. carpophila*, *P. manshurica*, and *H. opuntiae* in cocoa beans fermentation. | [143] |
| Fermentation | Addition of acetic acid bacterial culture starter in fermentation | Culture starter: 130 AAB from 3 countries (French Guiana, Ivory Coast, and Mexico). | The addition of *A. pasteurianus* starter culture increased the conversion of ethanol and lactic acid into acetoin or acetic acid, thereby improving quality of cocoa beans. | [84] |
| Fermentation | Addition of LAB culture starter in fermentation to inhibit the growth of fungi. | Starter LAB: *L. fermentum* and *L. plantarum*. Fermentation periods: 5 days. | The addition of *L. fermentum*, *L. plantarum*, and a combination of *L. plantarum* with *A. aceti* and *S. cerevisiae* provided suitable pH and temperature, and inhibited the growth of fungi. | [116] |
| Fermentation | Addition of indigenous LAB, yeast, AAB in fermentation to inhibit mycotoxins produced by fungi. | Starter: indigenous *Acetobacter* spp. *HA-37, L. plantarum HL-15, C. famata HY-37* at concentration of $10^9$ CFU/mL. Fermentation periods: 5 days. | The use of indigenous *L. plantarum HL-15* or in combination with *Acetobacter* spp. and *C. famata* inhibited ochratoxin-A produced by fungi. | [14] |
| Fermentation | Addition of BAL starter *L. plantarum* in the fermentation of cocoa beans. | Starter *L. plantarum*: $10^3$ CFU per gram cocoa beans. | The addition of BAL starter accelerated the growth of lactic acid bacteria and acetic acid bacteria. There was also an increase in the amount of acetic acid, lactic acid, and ethanol produced. Fermentation index increased and the time was shorter. | [151] |
| Fermentation and drying | Determining fermentation and drying times in the rainy season. | Fermentation periods: 5–8 days. Drying: 4–6 days. | Fermentation for 8 days and drying for 6 days using sunlight in the rainy season produced the best quality cocoa beans. | [27] |
| Drying | Drying cocoa beans with adsorption, vacuum drying, and freeze drying to get high antioxidant cocoa beans. | Pressure of freeze dryer: 0.015 mbar, condenser dimensions: 31.3 cm × 34.5 cm × 46.0 cm. | Freeze-drying produced cocoa beans with the highest antioxidant activity (71.8 mg Trolox/g) and polyphenol content (126.3 mg GAE/mg). | [157] |
| Drying | Drying cocoa beans with solar power equipped with a heat pump. | Temperature: 32–48 °C and RH: 35–80%. | A solar dryer with a heat pump can speed up the drying of cocoa beans from 6 days to 5 days. | [118] |

**Table 4.** *Cont.*

| Post-Harvest Stage | Treatments | Conditions | Characteristics | References |
|---|---|---|---|---|
| Storage | Re-fermentation of dry non-fermented cocoa beans by administering pure cultures to fermented the non-fermented of dry cocoa beans. | Moisture: 15%. Fermentation periods: 120 h. | The addition of pure culture of *A. aceti, L. lactis,* and *S. cerevisiae* improved quality of dry cocoa beans by facilitating the fermentation process and increasing the fermentation index up to 1.03. | [12] |

Fermentation modification involving various microbial starter cultures had been proven to improve the quality of cocoa beans, but the process was prone to contamination. Romanens et al. [156] reported that the spontaneous fermentation of cocoa beans led to the growth of mold, causing a decrease in quality. Controlled fermentation by administering two anti-fungi strains improved the anti-fungi activity in fermenting and drying to improve quality. The anti-mold cultures used included *S. cerevisiae H290* and *L. fermentum M017* cultures and *L. fermentum 223* and *S. cerevisiae H290 B* cultures. The dominance of *L. fermentum M017* and *223* caused an increase in mannitol originating from the ability of Lb. Culture B *L. fermentum 223* and *S. cerevisiae H290* showed the best results with less off-flavor, a good percentage of fermented beans, less astringency, and the best taste.

Several studies had also extensively explored fermentation modification, including Rahayu et al. [14], who added indigenous lactic acid bacteria, yeast, and acetic acid bacteria to inhibit mycotoxins produced by fungi. The use of indigenous *L. plantarum HL-15* or in combination with *Acetobacter* spp. and *C. famata* inhibited ochratoxin-A produced by fungi. Marwati et al. [116] also reported that the addition of *L. fermentum, L. plantarum,* and a combination of *L. plantarum* with *A. aceti* and *S. cerevisiae* provided suitable pH and temperature, as well as inhibited the growth of fungi.

Improving the quality of cocoa beans could also be done on non-fermented dry samples, which were known to be low quality. Improvements were made by administering pure cultures of *A. aceti, L. lactis,* and *S. cerevisiae* to fermented and non-fermented dry cocoa beans, as reported by Apriyanto et al. [12]. The results showed that fermentation treatment with the addition of pure culture gradually improved the quality of the samples. This was because the treatment facilitated fermentation and the role of microorganisms during the process could be controlled based on the succession of each microorganism.

The activity of microorganisms during post-harvest handling of cocoa beans had various beneficial effects, especially on fermentation. However, some risks or adverse effects could occur due to the activity of microorganisms. Some of the advantages and disadvantages of microorganisms' activity in the post-harvest handling of cocoa beans are presented in Table 5.

**Table 5.** The advantages and disadvantages of microorganisms' activity in post-harvest handling of cocoa beans.

| Advantages | | | |
|---|---|---|---|
| No. | Role | Microorganism | References |
| 1. | It decomposed pectin compounds into pectin acids and alcohol, and the pulp was crushed and released due to the decomposition of pectin. Sugars were then converted into alcohol compounds and citric acid was broken down. | Yeast | [25,75] |

**Table 5.** *Cont.*

| No. | Role | Microorganism | References |
|---|---|---|---|
| 2. | Broke down citric acid in cocoa fermentation sugar in the pulp through homofermentative and heterofermentative pathways. | Lactic acid bacteria | [75,105] |
| 3. | Played a role in the process of oxidation of alcohol compounds to acetic acid. | Acetic acid bacteria | [19,20] |
| 4. | Killed the seeds to ensure changes, such as the formation of color and flavor precursors in fermentation. | Acetic acid bacteria, yeast, and lactic acid bacteria | [158] |
| **Disadvantages** | | | |
| **No.** | **Role** | **Microorganism** | **References** |
| 1. | The growth of mold was a risk to public health due to the toxins it produced. | *Aspergillus, Penicillium,* and *Fusarium* | [159,160] |
| 2. | It caused typhus with symptoms of diarrhea, nausea, and dizziness if cocoa beans were contaminated. | *Salmonella* | [39] |
| 3. | The cause of weathering, reduced nutrition, and the presence of mycotoxins, which could cause health problems in cocoa beans. | Mold | [34] |
| 4. | Contamination by bacteria could cause nosocomial infections when cocoa pods were ripened. | *Acinetobacter* sp., *Klebsiella pneumoniae* | [57] |
| 5. | There was damage to the color and flavor of cocoa beans. | Kapang | [161] |

Some of the microorganisms' activities during post harvest were beneficial because they contributed to improving the quality of cocoa beans, including (1) yeast, which converted sugar into alcohol and citric acid; (2) lactic acid bacteria (LAB), which converted citric acid into lactic acid; and (3) acetic acid bacteria (AAB), which oxidized alcohol to acetic acid, which then played a role in killing the seeds, forming a brown color, and forming flavor precursor compounds. However, some microorganism activities could contaminate, which harmed or reduced the quality of cocoa beans, and were dangerous because they produced toxins. These microorganisms included (1) *Salmonella*, which caused diarrhea and typhus; (2) *Acinetobacter* sp. and *Klebsiella pneumoniae*, which caused nosocomial infections; and (3) *Aspergillus*, *Penicillium*, and *Fusarium*, which produced toxins that were harmful to health. Therefore, microbial contamination must be prevented and controlled during post harvest to maintain the quality of the cocoa beans.

*4.2. Technological Difficulties in Maintaining the Postharvest Quality of Cocoa Beans*

Post-harvest handling greatly affects the quality of the cocoa beans. However, many of the cocoa beans circulating in the market are of low quality due to improper post-harvest handling [162,163]. Various technological difficulties are still encountered in maintaining the post-harvest quality of cocoa beans, especially during the fermentation, drying, and storage processes. The process of fermenting cocoa beans takes quite a long time, around 5–6 days, causing some farmers not to ferment their cocoa beans or to ferment them

impatiently and incorrectly so that the quality of the cocoa beans is low. Until now, fermentation technology has been developed with various methods and various additions of microbial starters, but the fermentation time cannot be significantly shortened [84,116,156]. Shortened fermentation of cocoa beans can improve the activity of antioxidants and other bioactive compounds but cannot produce good flavor precursor compounds [52]. Dry cocoa beans that are not fermented are also a challenge because they are low quality. Several re-fermentation technologies have been applied to dry unfermented cocoa beans and can slightly improve the quality of cocoa beans, but not as good as cocoa beans fermented from scratch. This is because the substrate available for re-fermentation cannot support the type and number of microbial populations that should be involved in the fermentation process [12].

Drying technology is also still experiencing some difficulties. This is because the drying process of cocoa beans requires sufficient time to obtain a moisture content of less than 8% with good quality. Drying can be accelerated using various drying machines, but drying too fast can increase the acidity in the cocoa beans, while drying for too long can cause mold growth which can produce mycotoxins and reduce the quality of the cocoa beans [134]. Cocoa bean storage technology also has challenges in maintaining the quality of cocoa beans over a long period of time. Controlled atmosphere technology, such as nitrogen or carbon dioxide, is quite effective in maintaining the quality of cocoa beans, but it requires quite complicated installation and is expensive [139,140]. Using preservatives to prevent microbial growth also carries risks related to their toxicity and food safety. Therefore, it is still necessary to develop other storage innovations that are cheaper, more effective, and more efficient.

## 5. Conclusions and Future Research

Post-harvest handling of cocoa beans needs attention, starting from picking the pods, sorting, ripening the pods, fermenting, drying, and storing. The activity of microorganisms in each post-harvest stage affects the quality of cocoa beans. Fermentation is one of the important stages involving microorganisms such as acetic acid bacteria, lactic acid bacteria, and yeast which play a major role in producing precursor flavor compounds in cocoa beans. However, harmful microbes such as pathogenic bacteria and fungi can contaminate and reduce the quality of cocoa beans. Mold contamination causes weathering, reduced nutrition, and the presence of mycotoxins. Improvement of cocoa bean quality can be carried out by adding a starter culture in fermentation. The addition of these cultures can increase the fermentation rate and improve the biochemical process in cocoa beans. Proper drying processes and controlled storage can maintain the fermented cocoa beans' quality. However, many difficulties and challenges remain, especially those related to controlling microbial activity during post harvest. Post-harvest technology needs to be continuously developed, especially in controlling microbiological activities to improve the quality of cocoa beans.

Various studies to develop post-harvest handling technology had been carried out, especially to improve the quality of the product. Various fermentation techniques continued to be developed to create ideal conditions for microbiological activity. Various types of starters and various substrates were also added to the fermentation process to increase the speed and improve the quality of the products. Various efforts had also been made to improve the quality of non-fermented cocoa beans by further fermentation or by approaching the chemical compounds produced during the process using synthetic materials. Various technologies had also begun to be developed to increase the bioactive compounds in cocoa beans to provide high antioxidant activity, which was beneficial for health. Furthermore, several combinations of modification technologies were carried out to obtain quality cocoa beans, especially from the desired biochemical compound content and flavor precursors.

**Author Contributions:** Conceptualization, E.S. and E.L.; methodology, M.D. and N.B.; software, R.I.; validation, M.D., R.I. and E.L.; formal analysis, N.B.; investigation, E.S. and N.B.; resources, E.S. and N.B.; data curation, E.S. and N.B.; writing—original draft preparation, E.S. and N.B.; writing—review and editing, M.D., R.I. and E.L.; visualization, R.I.; supervision, M.D.; project administration, E.L.; funding acquisition, E.S. All authors have read and agreed to the published version of the manuscript.

**Funding:** This article was supported by Universitas Padjadjaran, Indonesia, grant number: 1549/UN6.3.1/ PT.00/2023.

**Data Availability Statement:** Not applicable.

**Acknowledgments:** Thank you to the Rector of Universitas Padjadjaran and The Ministry of Education, Culture, Research, and Technology of the Republic of Indonesia.

**Conflicts of Interest:** The authors declare no conflict of interest.

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
