# Peer review of "Microbiological Activity Affects Post-Harvest Quality of Cocoa (Theobroma cacao L.) Beans"

_horticulturae, doi:10.3390/horticulturae9070805_

Round 1

Reviewer 1 Report

Subroto et al.'s study provides a review of the microbiological aspects related to cocoa bean post-harvest. The authors examine the role of microorganisms in each step of the post-harvest process and emphasize their impact on chocolate quality and safety. While the topic is relevant and warrants investigation, there have been several recent reviews focusing on the microbiological characteristics of cocoa beans, particularly during fermentation. I believe the text does not bring new information to justify publication. I also have reservations about the current work due to its confusing and repetitive nature. The text frequently repeats words and ideas, which could be addressed through substantial editing. In its current format, I do not believe the manuscript is suitable for publication.

The writing style would greatly benefit from significant revisions, and the excessive repetition throughout the text should be minimized to enhance readability.

Author Response

Response to Reviewer 1 Comments

Subroto et al.'s study provides a review of the microbiological aspects related to cocoa bean post-harvest. The authors examine the role of microorganisms in each step of the post-harvest process and emphasize their impact on chocolate quality and safety. While the topic is relevant and warrants investigation, there have been several recent reviews focusing on the microbiological characteristics of cocoa beans, particularly during fermentation. I believe the text does not bring new information to justify publication. I also have reservations about the current work due to its confusing and repetitive nature. The text frequently repeats words and ideas, which could be addressed through substantial editing. In its current format, I do not believe the manuscript is suitable for publication.

The writing style would greatly benefit from significant revisions, and the excessive repetition throughout the text should be minimized to enhance readability.

Response:

The manuscript has been revised with substantial editing. The writing style has been revised, and the excessive repetition throughout the text has been minimized to enhance readability.

Reviewer 2 Report

Line 415 - 416 ==> The authors should refer to international standard instead

Line 417 ==> If there is already GMP standards for cocoa beans, the authors should include or cite it in this manuscript

Figure 3 ==> title of x-axis is missing. Use of patterns for the bar charts is more difficult to see than colors

Line 457 ==> Any recommendation of humidity cocoa beans should be stored?

Line 464 - 466 ==> The authors recommended the use of inert gases for storing cocoa beans, however the references are for other products. Is there any literatures or real life examples from using the inert gases for cocoa beans?

The quality of English language is acceptable.

Author Response

Response to Reviewer 2 Comments

Point 1: Line 415 - 416 ==> The authors should refer to international standard instead

Response 1:

The international standard for the Cocoa beans — Specification and quality requirements has been referred in the text (Page 14, Lines 422-425, and Page 29, Lines 994-995, red color).

Point 2: Line 417 ==> If there is already GMP standards for cocoa beans, the authors should include or cite it in this manuscript. 

Response 2:

GMP standards for cocoa beans have been included or cited in the manuscript (Page 14, Line 425-433, and Page 29, Lines 996-1001, red color).

Point 3: Figure 3 ==> title of x-axis is missing. Use of patterns for the bar charts is more difficult to see than colors.

Response 3:

The title of x-axis and information or caption for x-axis have been added in Figure 3 (Page 15, lines 451-454, red color).

Point 4: Line 457 ==> Any recommendation of humidity cocoa beans should be stored?

Response 4:

The recommendation of humidity cocoa beans has been added to the manuscript (Page 15, Lines 476-481, red color).

Point 5: Line 464 - 466 ==> The authors recommended the use of inert gases for storing cocoa beans, however the references are for other products. Is there any literatures or real life examples from using the inert gases for cocoa beans?

Response 5:

Literatures or real life examples from using the inert gases for cocoa beans have been added to the manuscript (Page 16, Line 486-488, and Page 30, Line 1031-1033, red color).

Reviewer 3 Report

The abstract needs a substantial revision. The aim of the study pointed out in the abstract is not clear and does not represent well the essence of the study. Furthermore, abstract do not contain specific conclusions.

The introduction is not structured properly and the information presented is very messy.

The number of bibliographic sources is adequate, more than 70% of the total bibliographic sources are from the last 5 years.

There are some grammatical errors and instances of badly worded/constructed sentences throughout the manuscript. Please refine the language carefully. 

Author Response

Response to Reviewer 3 Comments

Point 1: The abstract needs a substantial revision. The aim of the study pointed out in the abstract is not clear and does not represent well the essence of the study. Furthermore, abstract do not contain specific conclusions.

Response 1:

The abstract has been substantially revised. The aim of the study is pointed out in the abstract more clearly and well represents the essence of the study. Abstracts have also been added with specific conclusions. (Page 1, Line 9-25).

Point 2: The introduction is not structured properly and the information presented is very messy.

Response 2:

The introduction has been revised to be better structured, and the information presented is more systematic (Pages 1-2, Line 29-79, red color).

Point 3: The number of bibliographic sources is adequate, more than 70% of the total bibliographic sources are from the last 5 years.

Response 3:

Thank you.

Reviewer 4 Report

Dear Authors,

The paper  discusses in more depth the activities of microorganisms involved in the  post-harvest handling process of cocoa beans. This review also discusses the effects of the activities of these microorganisms on the quality, as well as efforts to improve the quality of cocoa beans, especially by adding starter culture in the fermentation process.

As such, the topic of the study is interesting. It is noteworthy, that the Authors showed diligence in the selection of references.  The indicated literature’s sources are very current.

However, there are major flaws in the study design, the conceptual framework and the writing of the manuscript that need to be fixed. The main concern of this manuscript relates to the “advance of thinking”. The presented review article needs to go beyond mere description and ‘state-of-the-literature’ summaries and develop the new ideas and ways of thinking.

Specific comments on the manuscript are as follows:

          Please, complete the text of the presented review in the aspect of pro and  postbiotics.

          Please, add the subsection about technological difficulties in maintaining the  postharvest quality of cocoa (Theobroma cacao L.) beans

          Please recheck thoroughly the whole article and improve its grammatical mistakes.

          All references should be cited by the same way.

          Recheck references according to the journal guidelines.

          The subsection “Conclusion” needs to be improved according to the main pointed concern.

From my standpoint, this manuscript can be considered for publication in Journal – Horticulturae,  after major revision,  given the above aspects.

Author Response

Response to Reviewer 4 Comments

The paper  discusses in more depth the activities of microorganisms involved in the  post-harvest handling process of cocoa beans. This review also discusses the effects of the activities of these microorganisms on the quality, as well as efforts to improve the quality of cocoa beans, especially by adding starter culture in the fermentation process.

As such, the topic of the study is interesting. It is noteworthy, that the Authors showed diligence in the selection of references.  The indicated literature’s sources are very current.

However, there are major flaws in the study design, the conceptual framework and the writing of the manuscript that need to be fixed. The main concern of this manuscript relates to the “advance of thinking”. The presented review article needs to go beyond mere description and ‘state-of-the-literature’ summaries and develop the new ideas and ways of thinking.

Specific comments on the manuscript are as follows:

Point 1: Please, complete the text of the presented review in the aspect of pro and postbiotics.

Response 1:

The aspect of pro and postbiotics in fermented cocoa beans has been added to the text (Page 12, Line 343-356).

Point 2: Please, add the subsection about technological difficulties in maintaining the postharvest quality of cocoa (Theobroma cacao L.) beans.

Response 2:

The subsection about technological difficulties in maintaining the post-harvest quality of cocoa has been added to the manuscript (Page 21, Line 625-654, red color).

Point 3: Please recheck thoroughly the whole article and improve its grammatical mistakes.

Response 3:

The whole article has been rechecked, and improved its grammatical mistakes.

Point 4: All references should be cited by the same way.

Response 4:

All references have been cited by the same way.

Point 5: Recheck references according to the journal guidelines.

Response 5:

References have been rechecked according to the journal guidelines (Pages 22-32, Lines 696-1092).

Point 6: The subsection “Conclusion” needs to be improved according to the main pointed concern.

Response 6:

The subsection “Conclusion” has been improved according to the main pointed concern (Page 21, Line 657-668, red color).

Round 2

Reviewer 1 Report

Extensive editing has been conducted on the text, resulting in a significant improvement compared to the previous version. In particular, the study conducted by Almeida et al. on microbial composition by culture independent methods and cross feeding in cocoa fermentation (https://doi.org/10.1016/j.foodres.2020.109034) could greatly enhance the discussion. Another work by this same author also deals with the microbiome of cocoa fermentation and should be considered, especially for integrating some of the tables that deal with microbial composition of cocoa fermentation in different countries - https://doi.org/10.1128/AEM.00584-21 

Some minor corrections throughout the text could still be made to further enhance its quality before final acceptance.

minor corrections

Author Response

Response to Reviewer 1 Comments (Round 2)

Extensive editing has been conducted on the text, resulting in a significant improvement compared to the previous version. In particular, the study conducted by Almeida et al. on microbial composition by culture independent methods and cross feeding in cocoa fermentation (https://doi.org/10.1016/j.foodres.2020.109034) could greatly enhance the discussion. Another work by this same author also deals with the microbiome of cocoa fermentation and should be considered, especially for integrating some of the tables that deal with microbial composition of cocoa fermentation in different countries - https://doi.org/10.1128/AEM.00584-21

Some minor corrections throughout the text could still be made to further enhance its quality before final acceptance.

Response:

The study conducted by Almeida et al. on microbial composition by culture independent methods and cross feeding in cocoa fermentation (https://doi.org/10.1016/j.foodres.2020.109034) has been added to the text (Page 6, Line 278-287, and page 27, Line 905-907, red color).

Another work by this same author which deals with the microbiome of cocoa fermentation https://doi.org/10.1128/AEM.00584-21 has been added to Table 2 which deals with the microbial composition of cocoa fermentation in different countries (Pages 10 and 11, and Page 28, Line 953-954, red color).

Reviewer 3 Report

The revised version of the paper can be published. The abstract has a clear aim. The introduction gives a nice and updated presentation of the main issues under research. Materials and methods allow the reproduction of the experiments. Results support the discussion, which is further supported by data from other authors. The conclusion, although very synthetic, can be accepted in the present stage.

Author Response

Response to Reviewer 3 Comments (Round 2)

The revised version of the paper can be published. The abstract has a clear aim. The introduction gives a nice and updated presentation of the main issues under research. Materials and methods allow the reproduction of the experiments. Results support the discussion, which is further supported by data from other authors. The conclusion, although very synthetic, can be accepted in the present stage.

Response:

Thank you.

Reviewer 4 Report

Dear Authors,

Thank you very much for the responses to the review.

The new version of  the manuscript has been corrected taking into account the previous aspects.

Only one aspect should be improved. You have to add current definition of probiotics, according to:

Hill, C.; Guarner, F.; Reid, G.; Gibson, G.R.; Merenstein, D.J.; Pot, B.; Morelli, L.; Canani, R.B.; Flint, H.J.; Salminen, S.; et al. (2014). The international scientific association for probiotics and prebiotics consensus statement on the scope and appropriate use of the term probiotic. Nat. Rev. Gastroenterol. Hepatol., 11, 506–514.

From my standpoint, this revised version of the manuscript will be appropriate for publication in Journal – Horticulture after minor revision,  given the above aspects.

Author Response

Response to Reviewer 4 Comments (Round 2)

Dear Authors,

Thank you very much for the responses to the review.

The new version of the manuscript has been corrected taking into account the previous aspects.

Only one aspect should be improved. You have to add current definition of probiotics, according to:

Hill, C.; Guarner, F.; Reid, G.; Gibson, G.R.; Merenstein, D.J.; Pot, B.; Morelli, L.; Canani, R.B.; Flint, H.J.; Salminen, S.; et al. (2014). The international scientific association for probiotics and prebiotics consensus statement on the scope and appropriate use of the term probiotic. Nat. Rev. Gastroenterol. Hepatol., 11, 506–514.

From my standpoint, this revised version of the manuscript will be appropriate for publication in Journal – Horticulture after minor revision,  given the above aspects.

Response:

The current definition of probiotics, according to Hill, C.; Guarner, F.; Reid, G.; Gibson, G.R.; Merenstein, D.J.; Pot, B.; Morelli, L.; Canani, R.B.; Flint, H.J.; Salminen, S.; et al. (2014). The international scientific association for probiotics and prebiotics consensus statement on the scope and appropriate use of the term probiotic. Nat. Rev. Gastroenterol. Hepatol., 11, 506–514. Has been added (Page 12, Line 353-354, and page 29, Line 970-972, red color).
